

# The Quasi-Liquid Layer of ice revisited: the role of temperature gradients and tip chemistry in AFM studies

Julián Gelman Constantin[1,2,#], Melisa M. Gianetti[2], María P. Longinotti[2], Horacio R. Corti[1,2]

[1] Departamento de Física de la Materia Condensada, Centro Atómico Constituyentes, Comisión Nacional de Energía Atómica, San Martín, B1650KNA, Buenos Aires, Argentina.
[2] Instituto de Química Física de los Materiales, Medio Ambiente y Energía (UBA-CONICET), Facultad de Ciencias Exactas y Naturales, Universidad de Buenos Aires, Pabellón II, Ciudad Universitaria, C1428EGA, Buenos Aires, Argentina.
# Current address: División de Química Atmosférica, Centro Atómico Constituyentes, Comisión Nacional de Energía Atómica, San Martín, B1650KNA, Buenos Aires, Argentina.

*Correspondence to*: Julián Gelman Constantin (juliangelman@cnea.gov.ar); María P. Longinotti (longinot@qi.fcen.uba.ar)

**Abstract.** In this work, we present new results of Atomic Force Microscopy (AFM) force curves over pure ice at different temperatures, performed with two different environmental chambers and different kind of AFM tips. Our results provide insight to resolve the controversy on the interpretation of experimental AFM curves on the ice-air interface for determining the thickness of the quasi-liquid layer (QLL). The use of a mini environmental chamber, that provides an accurate control of the temperature and humidity of the gases in contact with the sample, allowed us for the first time to get force curves over the ice-air interface without *jump-in* (jumps of the tip onto the ice surface, widely observed in previous studies). These results suggest a QLL thickness below 1 nm within the explored temperature range (-7 ℃ to -2 ℃). This upper bound is significantly lower than most of the previous AFM results, which suggests that previous authors overestimate the equilibrium QLL thickness, due to temperature gradients, or indentation of ice during the jump-in. Additionally, we proved that the hydrophobicity of AFM tips affects significantly the results of the experiments. Overall, this work shows that, if one chooses properly the experimental conditions, the QLL thicknesses obtained by AFM lay over the lower bound of the highly disperse results reported in the literature. This allows estimating upper boundaries for the QLL thicknesses, which is relevant to validate QLL theories, and to improve multiphase atmospheric chemistry models.

## 1. Introduction

Slightly below the melting temperature, $T_m$, surface premelting has been observed in many crystalline solids. This melted layer is commonly called in the literature "quasi-liquid layer" (QLL), since many of its properties differ from those corresponding to the bulk supercooled liquid at the same temperature. The existence of a QLL at the ice-air interface has been thoroughly discussed in the literature, mainly considering that this layer plays an important role in the flow behavior of ice and snow, the adsorption of substances onto ice, and the low friction of solids on ice (Petrenko, 1994; Wettlaufer and Dash, 2000; Anderson and Neff, 2008).





The relevance of the QLL in the atmospheric chemistry of clouds, polar regions, glaciers, and other cold regions is paramount, and it has been widely discussed in the literature.

As an example, Molina and coworkers (McNeill et al. 2007) studied the interaction of HCl with polar stratospheric cloud ice particles and found that the solute can induce the formation of a QLL at the characteristic temperatures of these clouds.

Grannas et al. (Grannas et al, 2007) emphasized the need of describing the chemistry occurring inside the QLL for modeling the snow photochemistry. Following this line, Boxe and Saiz-Lopez (Boxe and Saiz-Lopez, 2008) developed a multiphase model (CON-AIR) to deal with the condensed phase chemistry and photochemistry in the QLL, and applied it to the photochemistry of nitrate ($NO_3^-$), in the Artic and coastal Antartic snowpack.

The model developed by Kuo et al. (Kuo et al., 2011) follow a different approach and focuses on the formation of a brine
layer (BL) as a consequence of freezing of aqueous solutions with high solute contents. The authors emphasized that under relatively pristine conditions for which a brine layer is not predicted, a quasi-liquid layer may still be present and can significantly affect interfacial chemistry.

The physics of the premelting phenomena in ice and its geophysical consequences have been reviewed by Dash et al. (Dash et al, 2006) and Bartels-Rausch et al. (Bartels-Rausch et al, 2014), who reported a comparison between calculated and
measured QLL thicknesses.

Measurements of the ice QLL layer thickness were reported in the literature using different experimental techniques such as Brewster reflectometry (Elbaum et al., 1993), ellipsometry (Beaglehole and Nason, 1980; Furukawa et al., 1987), X-Ray scattering (Lied et al., 1994; Dosch et al., 1995 and 1996), proton channeling (Golecki and Jaccard, 1977), nuclear magnetic resonance (NMR) (Ishizaki et al., 1996), infrared spectroscopy (Sadtchenko and Ewing, 2002 and 2003; Richardson, 2006),
photoelectron spectroscopy (XPS) (Bluhm et al., 2002), and atomic force microscopy (AFM) (Petrenko, 1997; Bluhm and Salmeron, 1999; Bluhm et al., 2000; Döppenschmidt and Butt, 2000; Pittenger et al., 2001).

Molecular dynamic simulation results also demonstrate the existence of a QLL on the ice-air interface (Weber and Stillinger, 1983; Kroes, 1992; Furukawa and Nada, 1997; Limmer and Chandler, 2002; Carignano, 2007; Conde et al, 2008), and some of those works estimate its thickness. Figure 1 summarizes experimental and simulation results for the QLL thickness as a
function of the supercooling degree. As it can be observed in this Figure, simulations predict smaller QLL thicknesses than all experimental methods. Among the experimental techniques, the smaller QLL thicknesses correspond to measurements such as Brewster reflectometry (Elbaum et al., 1993), XPS (Bluhm et al, 2002), X-ray scattering (Lied et al., 1994; Dosch et al., 1995 and 1996) and IR spectroscopy techniques (Sadtchenko and Ewing, 2002 and 2003). On the contrary, ellipsometry (Beaglehole and Nason, 1980; Furukawa et al., 1987) and AFM determinations (Petrenko, 1997; Bluhm and Salmeron, 1999;
Döppenschmidt and Butt, 2000; Bluhm et al., 2000; Pittenger et al., 2001) give thicker QLL values. AFM experiments, for instance, involve the interaction of a tip with the sample; thus, it is uncertain whether other phenomena are also involved in the measurements.

Some of the QLL thickness determinations using AFM were performed by analyzing force curves (Petrenko, 1997; Döppenschmidt and Butt, 2000; Pittenger et al., 2001); that is, measuring the force experienced by the AFM tip as it





approaches the ice. While the tip is away from the surface the force between the tip and the sample is zero. At a certain point, close to the surface, the tip jumps into it, experiencing a negative force. The distance tip-surface at which this occurs is called jump-in distance, and in some cases is interpreted as the QLL thickness (Petrenko, 1997; Döppenschmidt and Butt, 2000), while some corrections were proposed by several authors. Distinctively, Bluhm and Salmeron (Bluhm and Salmeron,

1999) analyzed the QLL thickness by comparing AFM contact and non-contact experiments.

Petrenko (Petrenko, 1997) found difficult to explore the ice-air interface and get reproducible force curve measurements, especially due to adhesion of the AFM tips to the ice surface. Some of these measurements were thus performed by depositing a drop of decane above the ice surface in order to overcome these complications. In this work the author compared the time of interaction of the tip with the ice ($\tau_{int}$), with the time required to establish thermal equilibrium in the

contact point ($\tau_{eq}$) and estimated $\tau_{eq} \approx 3$ ns and $\tau_{int}$ around 50 ms, and concluded that the experiments give more information on the tip-ice or tip-QLL interface that on the ice-air interface. Nevertheless, the QLL thickness for the ice-air interface for one temperature was reported, as well as an estimation of the tip-QLL interfacial energy.

Pittenger et al. (Pittenger et al., 2001) analyzed the force curves between the tip and the ice for different indentation/penetration rates. The ratio between force and indentation rate was studied using silicon tips, with and without

hydrophobic coating, and evidence of the presence of a QLL between the tip and the ice was found between -1 and -10 ℃, based on the observation that the mentioned ratio is constant for a given pit depth. However, below -10 ℃, the dependence of the force with indentation rate changes, suggesting that plastic flow of the ice dominates. In addition, Pittenger et al. used a simple model for the viscous flow and observed that, to explain the experimental results, the viscosity of the ice-tip QLL should be considerably higher than that for supercooled water. Another important observation by these authors is that the

thickness of the QLL at the ice-tip interface depends on the hydrophobicity of the tips, obtaining smaller values for hydrophobic tips. Regarding the QLL at the ice-air interface, the authors question the ability of AFM experiments to properly measure this thickness, as will be discussed in more detail in the Results section.

Recent Molecular Dynamics simulations by Gelman Constantin et al. (Gelman Constantin et al., 2015a) show that hydrophobic tips promote the presence of a QLL at the tip-ice interface during indentation, supporting the analysis

performed by Pittenger et al. (Pittenger et al., 2001). The authors do not find an effect of the tip on the thickness of the QLL on the ice-air interface. However, further simulations by Gelman Constantin (Gelman Constantin, 2015b) show that hydrophilic tips can induce frustrated capillarity between the tip and the QLL. These results may explain the attractive interactions between ice-air interfaces and AFM tips (i.e., the jump-in). Additionally, the results show that during the approach of the tip, the QLL may deform to reach the tip due to a frustrated capillary (Goertz et al, 2009), leading to an

artificially enhanced QLL thickness result. Even though the semi-empirical potentials used in these simulations need further validation, these results support the hypothesis that the hydrophilicity of the tip may modify the measured QLL thickness.

Considering the high dispersion in the AFM QLL thickness values reported in the literature it is of fundamental relevance to analyze which factors may be involved in this dispersion of the data. For instance, the hydrophilicity of the tips was considered to influence the thickness results, while no quantitative measurements of the influence of the temperature



gradients in the air in contact with the ice sample where reported in the literature. A systematic study on other factors that which could possibly affect these determinations (size of the tips, speed of the force curves) is out of the scope of this article. Nevertheless, it should be noted that we choose the smallest available tip sizes, in order to avoid possible artifacts with larger tips, as in IFM studies (Goertz et al., 2009). The explored speed of the force curves, which is not informed, did not affect our

measurements on QLL thickness, while it does have an effect on ice indentation, which is not detailed in this study (Gelman Constantin, 2015b).

In the present work we critically analyze previous experimental AFM results in comparison to our new results. We measured force curves between the ice-air interface and AFM tips of varying hydrophilicity, with special care in reducing temperature gradients at the ice-air interface. This study shows that the jump-in distances obtained from the AFM force curves are very

sensitive to temperature gradients and tip hydrophilicity. Our new results for the QLL thickness are analyzed and compared to those reported using other experimental techniques. The discussion is focused in solving previous controversies on how to determine the thickness of the QLL using AFM technique and how the results could be compared with those obtained using other techniques, and establishing reasonable criteria for limiting the large scatter of data previously observed for this important parameter.

**2. Methods**

**2.1. Atomic Force Microscopy measurements**

QLL thickness measurements were performed with a commercial Atomic Force Microscope (AFM) by Veeco (currently Bruker), model Multimode, using the NanoScopeIIIa controller and the Quadrex module. Force curves were registered at a frequency of 1.744 Hz and a sampling of 8192 data points per curve. Measurements were performed with different

commercial AFM tips provided by Bruker (silicon, silicon nitride, and Pt/Ir coated silicon), whose characteristics are summarized in Table S1 of the Supplement. Additionally, we used a commercial silicon nitride tip functionalized by immersion in 1 M chlorotrimethylsilane in heptane.

Raw data generated by the AFM operating software (NanoScope 5.30r3, Veeco) was exported with NanoScope Analysis 1.40 (Bruker) and post-processed with an in-house developed software written in Scilab 5.4.0 (Scilab Enterprises, 2012). Our

software allows a semi-automatic analysis of force curves and saves individual image files, showing the shape of the curves and the regions analyzed, and a spreadsheet with the quantitative information extracted from the curves.

Calibrations required for a proper analysis of the AFM force curves are detailed in the Supplement.

**2.2. Characterization of AFM tips by EDS**

Pt/Ir coated silicon AFM tips (SCM-PIC, Table S1) were characterized by Scanning Electron Microscopy with Energy

Dispersive XR Spectroscopy (Carl Zeiss NTS SUPRA 40 at Centro de Microscopías Avanzadas, Facultad de Ciencias





Exactas y Naturales, Universidad de Buenos Aires). The goal of this characterization is to compare the morphology and composition of the tips before and after usage in AFM indentation experiments.

### 2.3. Humidity and temperature control

The temperature of the sample at the AFM was controlled with a set of commercial accessories provided by Veeco: the Thermal Applications Controller, the Sample Heater/Cooler Peltier, and the Heater/Cooler Scanner HC-AS-130V. This allows controlling the sample temperature between -30 ℃ and 100 ℃ with a precision of ± 0.1 ℃. The relative humidity (RH) and temperature of air in contact with the sample was measured with a Sensirion humidity-temperature sensor (model SH71).

Two environmental chambers, which will be further described in the following paragraphs, were developed in this work to control the RH of the sample and, in one case, the temperature of the air in contact with the ice.

### 2.3.1. Environmental Chamber (EC).

The in-house developed Environmental Chamber (EC) is composed by two main elements, as shown in Fig. 2: an acrylic chamber (A), and an aluminum ring (F). The top side of the aluminum ring has a thread and a groove for an o-ring for the sealing with the acrylic chamber (A), while the bottom side has a groove for an o-ring that completes the seal with the AFM base (E). Additionally, the ring has several sealed connections that allow gases inlet and outlet (D), electrical connections, and humidity and temperature sensors. Humidity control within this chamber was performed by mixing dry nitrogen (Indura 4.8, with less than 3 ppm of $H_2O$) with water saturated nitrogen. This chamber has some minor gas leaks (due to constraints imposed by the design of the AFM base), so all measurements had to be performed under continuous gas flow. We circulated between 1 and 6 $dm^3$/min of dry nitrogen (measured with a flow meter Argenflow, calibrated between 1 and 10 $dm^3$/min), and between 10 and 200 $cm^3$/min of water saturated nitrogen (measured with an Alicat flow controller having a maximum flow rate of 200 $cm^3$/min).

This chamber allows a good control of humidity, but it does not provide control of the temperature of the gases in contact with the sample. Additionally, the acrylic chamber makes impossible the access to the laser beam and detector adjustment screws. This is crucial because sometimes both laser's alignment and detector's position need to be adjusted during measurement. Hence, as we will show below, we find that our second version (the Mini Environmental Chamber) is a much better accessory to study this kind of systems. Nevertheless, a comparison between the results obtained with both chambers allows gaining new insight on the ice-air interface.

### 2.3.2. Mini Environmental Chamber (mEC).

The Mini Environmental Chamber (mEC) was designed to reduce the volume of air in contact with the sample, where air humidity and temperature must be controlled. The main element of the mEC is an AFM glass fluid-cell (F in Fig. 3). This cell has holes for the inlet (J) and outlet of gases (I), and to locate a humidity-temperature sensor (in the outlet of gases, I). It





also has a groove in the bottom face for a silicon o-ring (D), that seals the space between the cell and the substrate (C). Humidity control in this space was performed in a similar way than in the EC, but with much lower flow rates (we used two Alicat flow controllers, with maximum flow rates of 200 cm$^3$/min and 50 cm$^3$/min, for dry and humid airs, respectively).

In order to control the temperature of the gases in contact with the sample, we designed a copper cooler (G in Fig. 3), for
circulation of cold nitrogen vapor. Liquid nitrogen flows by siphon effect from an insulated flask, by the overpressure due to its own evaporation. The flow rate is controlled by a vent valve in the insulated flask that controls the overpressure. Cold nitrogen vapor, generated by evaporation of the liquid by contact of the tubes with air at room temperature, reaches the glass fluid cell (F in Fig. 3). The fine temperature control was achieved with an in-house developed heater (H in Fig. 3), made with Nichrome (nickel-chromium alloy) wire coiled around a mica sheet, and electrically isolated with an additional mica sheet
on the bottom side, and a glass slide on the top side. The total heater resistance was around 18 $\Omega$. The heater was powered by a PID controller (TERMOLD, NG-2 model), which measured the temperature at the copper cooler with a platinum resistance sensor (Honeywell HEL-777-A-T-0, 100). This system allows controlling the temperature of the copper cooler with fluctuations below $\pm$ 0.5 ºC.

### 2.4. Ice samples preparation

Ice samples to be measured at a working temperature ($T_{ice}$) (controlled and measured with the Peltier accessory below the sample) were prepared by controlled vapor deposition using the following procedure:

1.  The humidity in the EC or mEC was maintained below 80% RH (relative to $T_{ice}$) using dry nitrogen gas, in order to avoid ice or water condensation during calibration.
2.  When using the mEC, the copper cooler temperature ($T_{cooler}$) was set between 3 and 6 K above $T_{ice}$. This temperature gradient could not be further reduced, since for $T_{cooler}$ closer to $T_{ice}$ we observed condensation on the fluid cell and/or on the AFM tip, as we will discuss in the following sections.
3.  After calibration, we first controlled the desired RH, between 90% and 105% (relative to $T_{ice}$), while keeping the mica substrate temperature ($T_s$) 2 to 5 K above the desired measuring temperature ($T_{ice}$) to avoid condensation due
to RH fluctuations that may occur during this step. Oversaturation conditions made more difficult to measure contact images or force curves on the ice-QLL surface (due to condensation on the tip). Hence, in most of the experiments we worked at slight under-saturation conditions.
4.  By using the Peltier accessory, $T_s$ was then lowered 2 to 3 K below $T_{ice}$, reaching oversaturation (RH between 105% and 120%). Ice deposition was then allowed during 4 to 8 minutes. It must be noted that inverting steps 3 and 4 is
not preferred. In order to obtain reproducible, equilibrium ice layers, one should increase oversaturation slowly and steadily. This can be achieved more easily lowering $T_s$ with the commercial accessory other than increasing humidity.



5. The temperature was raised from $T_s$ to $T_{ice}$, and the ice was stabilized for 10 to 20 minutes prior to measure the force curves.

The small volume of the mEC allowed observing the RH changes during the ice deposition protocol. Figure 4 shows the RH changes during the sample preparation, where a marked drop of humidity almost immediately after the drop in temperature

(due to ice deposition) can be observed. When temperature raises again, humidity increases too, and RH reaches values close to 100% (relative to $T_{ice}$).

Another set of experiments were performed in the mEC during deposition of ice at a RH around 120%. These assays allow comparing the QLL thickness obtained at RH around 100%, for a stabilized ice sample, with those obtained upon oversaturation and a non-stabilized ice sample.

**2.5. Preparation of thin layers of glycerol**

In order to evaluate the hydrophilicity of AFM tips, we prepared thin layers of glycerol on glass slides and silicon wafers. The substrates were treated with Piranha solution (3:1 mixture of concentrated sulfuric acid and 30% hydrogen peroxide solution) in order to remove organic impurities and increase hydrophilicity of the surface. We prepared the glycerol films by spin-coating at different velocities, obtaining metastable layers less than 5 µm in thickness (as measured with AFM force

curves).

**3. Results and Discussion**

**3.1. AFM measurements with the EC**

Force curves on ice obtained with the EC with silicon tips in the temperature range from -7 ℃ to -2 ℃ were similar to those previously described (Petrenko, 1997; Butt et al., 2000; Döppenschmidt and Butt, 2000; Pittenger et al., 2001). Their general

shape can be seen in the central panel of Fig. 5, as compared with the force curve determined on mica (top panel). As discussed in Introduction and in the Supplement, the usual interpretation of this kind of force curve is that the jump-in distance is related to the interaction with the ice interface prior to contact (possibly a frustrated capillarity with a QLL), and that contact with the solid ice surface begins at $z_{tip} = 0$ and extends in the lineal region that follows (positive $z_{tip}$ values). Indentation slopes in ice are in the same order of magnitude to those found in other works in similar conditions, as they

depend on temperature, velocity and tip shape (Pittenger et al., 2001, Fig. 4).

Table 1 reports relevant features of the measured force curves. Force curves measured on the same position over the ice surface show high reproducibility, as previously reported (Petrenko, 1997; Butt et al., 2001; Pittenger et al., 2001). Indentation slopes show standard deviations lower than 1% (it must be noticed that the significant figures reported are consistent with the propagated uncertainty, which is higher, due to the uncertainty of the spring constant). Jump-in distances

show higher standard deviations (around 10%). Such reproducibility, also reported by Petrenko (Petrenko, 1997), suggests



that after indentation, the ice surface reconstructs quickly by capillary condensation (Pittenger et al., 2001) or flow from the QLL.

Some authors have assigned the observed jump-in distance to the thickness of the QLL (Petrenko, 1997; Döppenschmidt and Butt, 2000), on the basis of two assumptions: 1) the jump-in starts just when the tip makes contact with the QLL; 2) the

jump-in ends when the tip reaches the solid layer beneath the QLL. These assumptions have been under discussion in the community (Döppenschmidt and Butt, 2000; Pittenger et al., 2001). Regarding the first assumption, Doppenschmidt and Butt apply a small correction on the jump-in distance (from 1 to 2 nm) considering van der Waals forces. Mate et al. (Mate et al., 1989) estimate a larger bias, around 7 nm, if the tip is covered by a liquid film prior to contact when measuring the thickness of a 22 nm liquid film. Computer simulations (Gelman Constantin, 2015) have also shown that for hydrophilic AFM tips the

QLL deforms to reach the tip (frustrated capillarity), which would produce jump-in distances larger than equilibrium QLL thicknesses. The second assumption has been questioned as well, and will be discussed in more detail in this section.

Jump-in distances obtained in our EC are reported in Table 1 and plotted in Fig. 6 together with results from similar studies (Petrenko, 1997; Bluhm et al., 1999 and 2000; Döppenschmidt and Butt, 2000; Pittenger et al., 2001; Goertz et al., 2009). Symbols for the same temperature represent measurements on different positions over the ice surface. Error bars correspond

to twice the standard deviation of several measurements performed at a fixed position. The set of measurements exhibit a weak dependence with temperature, which would become much weaker if the measurement corresponding to the largest jump-in at $T_m$-$T$ = 2.0 K would be discarded. The dispersion of the measurements at the same temperature may have different explanations. For instance, temperature gradients in the sample could produce differences in QLL thicknesses or a patch of mica with no solid water and only a liquid layer can exist. The effect of temperature gradients will be quantitatively

discussed in the section 3.2.

A first order comparison with the literature shows that our QLL thickness results lie in the lower range of all reported values. QLL thicknesses reported by Pittenger et al. (Pittenger et al., 2001) are in the same range that ours, whereas Döppenschmidt and Butt (Döppenschmidt and Butt, 2000) reported higher values, with a larger temperature dependence. Pittenger et al. (Pittenger et al., 2001) state that the jump-in distances determined in their experiments are not representative of the QLL

thickness, since they are almost constant (around 3 nm) over a wide temperature range (1-17 K below the melting temperature). They also doubt on the reliability of the large QLL thicknesses measured by Döppenschmidt et al. assuming that the tip penetrates the ice during the jump-in. Hence, the above mentioned assumption that the jump-in ends when the tip reaches the solid layer beneath the QLL, will not be valid. In fact, they show that ice indentation distance at zero force has a strong dependence on temperature (as we found for experiments in the mEC, as we will discuss next).

Computer simulations (Gelman Constantin et al., 2015; Gelman Constantin, 2015) suggest that hydrophilic tips indent ice more easily (with lower free energy barriers, due to the attractive interaction) than their hydrophobic analogues. In addition, it was found that the ice layers below the QLL deform prior to contact with the tip. These results also reinforce the hypothesis by Pittenger et al. (Pittenger et al., 2001) that QLL thicknesses obtained from AFM jump-in distances with hydrophilic tips are overestimated due to ice indentation. It should be stressed that the same artifact could affect Petrenko's



results, although the author uses stiffer cantilevers (which reduce the ice indentation distance, since it compensates capillary forces at smaller deflections). In addition, Petrenko obtained the deflection sensitivity from the retract portion of force curves on ice at low temperatures, instead of using a more rigid substrate like mica or glass, as we did here. Therefore, the procedure adopted by Petrenko could lead to an underestimation of the deflection sensitivity (Attard, 2007), which would

imply an overestimation of the jump-in distances.

Results by Bluhm et al. (Bluhm and Salmeron, 1999; Bluhm et al., 2000) present an opposite trend to most of the literature values, that is, lower QLL thicknesses at higher temperatures. However, it should be noted that they measure QLL thickness over much thinner layers of ice (0.3 nm to 3 nm), where the QLL and the ice thicknesses are comparable.

Goertz et al. (Goertz et al., 2009) obtained QLL thickness much higher than the AFM results previously discussed, by using

Interfacial Force Microscopy (IFM). The experiment is very similar to AFM force curves, but the setup avoids mechanical instabilities (jump-in and pull-off) and generally uses larger tips. The difference between IFM and AFM results could be due to the difference in the size of the tip (150 µm spherical tip radius), to the poor control of the sample temperature (temperature is only controlled below the sample, while the large glass tip was not cooled), or to a fail in the suppositions required for the analysis of the force curves, as described previously.

**3.2 AFM measurements with the mEC**

AFM force curves measurements with the mEC using silicon tips (SNL, as in the experiments in the EC) or silicon nitride tips (DNP) over the same temperature range, whose results are summarized in Table 2, exhibit some differences with those obtained with the EC.

Firstly, with both kinds of tips, many experiments (seven out of twelve) ended with a complete loss of the signal in the laser

detector. We found two possible explanations for these observations. This new configuration (mEC), with lower temperature gradients in the chamber, surely leads to lower temperature on the tip. This may allow some condensation on the reflecting back coating of the cantilever, which prevents the laser beam from reaching the detector. Moreover, the reduction of the temperature gradients could also have an effect on the attractive interactions between the tip and the ice surface due to condensation on the tip, or changes on the ice surface. An increase of the attractive interactions could lead to a large bending

of the cantilever, with a consequent large deflection of the laser spot out of the area of the detector.

Secondly, some other experiments (four of them) produced force curves with no jump-in (like in the lower panel in Fig. 5). The absence of jump-in led us to suggest that in those experiments, the QLL, if present, should have a thickness lower than 1 nm (the minimum jump-in distance that could be detected in these experiments, due to noise in the deflection signal). The force curves indicate that the tip goes abruptly from the vapor phase (horizontal region, no net forces on the tip) to indenting

the solid ice phase (diagonal linear region). Indentation slopes are reported in Table 2; a comparison with those reported in Table 1 is out of the scope of this article and needs to take into the account the tip shape and the uncertainty of the spring constant and that of deflection sensitivity. Thus, how can we explain the difference with the results using the EC? Temperature gradients along the sample in the EC could be the key.




If the temperature of the air layer in contact with the ice surface is higher than $T_m$, a liquid layer can cover the sample. A simple heat transfer calculation (chapter 11, Bird et al., 2007) can provide the thickness of the stationary state liquid layer:

$$S(t \to \infty) = L \left(1 - \frac{T_L \kappa_l}{T_0 \kappa_s}\right)^{-1} \qquad (1)$$

where $S$ is the thickness of the liquid layer, L is the total thickness of the system (ice + liquid water), $T_L$ and $T_0$ are the temperature of the system surface (in contact with air) and the ice bottom (in contact with the Peltier element), respectively, and $\kappa_l$ and $\kappa_s$ are the thermal conductivities of liquid and solid phases. It is evident, from the results displayed in Fig. 7, that if the air in contact with the sample is only slightly above 0 ºC, thick liquid layers appear over the ice surface. Clearly, this will

lead to an overestimation of the QLL thickness. Smaller temperature gradients (where ice surface presents a temperature larger than reported, but below 0 ºC) could also affect QLL thickness measurements. It must be noted that among the experimental techniques reviewed in this article, the smaller QLL thicknesses measurements were achieved in experiments in which ice was only in contact with water vapor, such as Brewster reflectometry (Elbaum et al., 1993), XPS (Bluhm et al, 2002), X-ray scattering (Lied et al., 1994; Dosch et al., 1995 and 1996) and IR spectroscopy techniques (Sadtchenko and

Ewing, 2002 and 2003). For these measurements below 0 ºC, the low thermal conductivity in the gas phase implies lower heat transfer between the gases and the QLL, which can be easily compensated by the cooling system, resulting in small temperature gradients through the sample, and yielding to smaller QLL thicknesses. On the contrary, ellipsometry (Beaglehole and Nason, 1980; Furukawa et al., 1987) and AFM determinations (Petrenko, 1997; Bluhm and Salmeron, 1999; Döppenschmidt and Butt, 2000; Bluhm et al., 2000; Pittenger et al., 2001) give thicker QLL values. Even though in the

ellipsometry experiments the samples were only in contact with water vapor, Beaglehole and Nason and Furukawa et al. experiments use thick ice samples (several millimeters to centimeters) that are only cooled from below, which might produce relevant temperature gradients in the samples. For instance, in one of the experiments (Beaglehole and Nason, 1980) the authors report temperature gradients around 0.5 ºC in the ice sample.

For one of the experiments with silicon nitride tips (DNP1) using the mEC, we performed force curves at the same $T_{ice}$ but

adjusting $T_{cooler}$ at several temperatures. Table 2 shows the results for two of such $T_{cooler}$ temperatures and $T_{ice}$ = -5 ºC. Jump-in distances do not show a significant difference for both temperatures, although it seems that for higher $T_{cooler}$ the distances tend to be slightly higher and with a higher dispersion. However, there is a clear difference in the indentation slope: the lower $T_{cooler}$, the steeper the indentation slope. In other words, when the temperature gradient is higher, it is easier for the tip to indent the first layers of ice. Even though we could not extend the measurements with the mEC to the range of

temperature gradients that exist in the experiments with the EC (where $T_{cooler}$ could be considered close to ambient temperature), one may suppose that the effect on the slope should be even higher. This observation is consistent with that by Pittenger et al. (Pittenger, 2001) on the temperature dependence of the indentation distance at zero force, and supports his





claim that QLL thicknesses obtained from AFM jump-in distances (Petrenko, 1997; Döppenschmidt and Butt, 2000) are overestimated, as discussed above.

Figures S3 and S4 of the Supplement show that the force curves determined in the mEC with silicon nitride tips, as it was previously mentioned for the EC, are very reproducible. The reproducibility of all the force curves can be captured by

considering the averages and standard deviations of ice indentation slopes and jump-in distances on Table 2.

It can be stressed that experiments with more hydrophobic tips could be more appropriate for these studies. Petrenko (Petrenko, 1997) noticed that silicon tips were less appropriate than more hydrophobic tips to study the ice-QLL interface, due to high adhesion forces. With lower adhesion forces, the biases that affect the jump-in distances due to capillary forces would be smaller. Firstly, because using hydrophobic tips reduces the deformation of the QLL and the tendency to form a

neck between the tip and the sample. Secondly, because it reduces the net attractive forces acting on the tip, which produces a lower indentation of the solid ice during the jump-in. Hence, we performed further experiments with four platinum-covered tips and a silicon nitride tip functionalized with trimethylchlorosilane. In all cases, we obtained curves with no jump-in (type *c* curves in Fig. 5). Figure S5 of the Supplement shows as mode of example some force curves determined with Pt coated tips for the approach and retract branches of the curves, where it can be observed that no adhesion forces are present in the

retract portions of the curves.

In two cases (PIC1 and DNP-S1, Table 2), after repeated use of the tips, we started to obtain different force curves, similar to type *b* in Fig. 5. We believe that adhesion between the tip and the sample lead to detachment of the covering layer in those tips, increasing the overall hydrophilicity of the tip, and causing the artifices described above for silicon and silicon nitride tips. Figure 8 shows the SEM micrographs of the AFM Pt/Ir coated silicon AFM tips, before and after indentation

experiments, where it can be observed that, after usage, the roughness of the AFM tip increases markedly. Table 3 shows the EDS results obtained for the AFM tips before and after usage. Results show that after usage the relative amount of Pt in the sample decreases, probably due to a loss of the Pt cover in the tip zone that indents the ice. It should be stressed that Pt content does not reduce to zero. This is probably due to the fact that the EDS experiments collect X-ray radiation that originates in a volume of approximately 1 $\mu m^3$ around the AFM tip apex, which is larger than the indentation volume. Hence,

results in Table 3 are probably affected by a region of the tip not in contact with the sample, which may keep its Pt coating. It can also be observed, that S, Cl and Zn appear after indentation experiments, probably due to contamination of the tip with residues of these elements in the mica sheet.

Another set of experiments was performed using platinum covered AFM tips during deposition of ice at -5 ºC and 110 % RH (with tip PIC6) and at -2.5 ºC and 113 % RH (with tip PIC4). Both experiments show force curves with jump-in (similar to

those in the central panel of Fig. 5). QLL thickness values obtained for these experiments are 11.8 ± 3.3 nm for $T_{ice}$ = -5.0 ºC, $T_{cooler}$= -4 to -6 ºC and RH 110 %; and 23.8 ± 4.1 nm for $T_{ice}$ = -2.5 ºC, $T_{cooler}$ = 5 to 6 ºC and RH 113 %. These results are similar to those obtained with the more hydrophilic tips, probably because they were obtained upon repeated indentations of ice samples. Thus, the platinum coverage on the tip apex was probably lost during the measurements, as mentioned for some of the experiments previously described. Additionally, it should be stressed that in the later experiments force curves were



determined during deposition of ice under oversaturation conditions, whereas all our previous results correspond to slight undersaturation conditions where ice was allowed to stabilize 10-20 minutes prior to measurements. Thus, the later experimental conditions could lead to thicker (non-equilibrium) QLL values. Paleico et. al (Paleico et al., 2015) performed Grand-Canonical MD simulations under condensation conditions (over-saturation) and found non-equilibrium QLL

thicknesses larger than the equilibrium values.

Summarizing, the absence of a jump-in on some of the force curves measured in this work (more systematically with hydrophobic tips in the mEC) enforces the hypothesis that the QLL thickness is below 1 nm in the temperature range of the experiments (-7 ℃ to -2 ℃). However, a doubt arises: could it be the case that the later tips are not hydrophilic enough and hence the capillary force with the QLL (if present) is not strong enough to cause a jump-in? To discard this possible artifact

of our experiments, we studied the hydrophilicity of the more hydrophobic tips used. We used a spin coater to generate thin layers of glycerol over glass and silicon slides. We used glycerol instead of water due to the low vapor pressure of glycerol, which allows generating thin liquid films of approximately constant thickness. Additionally, glycerol is more viscous than water, and hence is a better probe of the QLL (which is expected to be more viscous than bulk liquid water) (Goetz et al., 2009). Figure 9 represents a force curve obtained with one of the platinum covered tips over a thin glycerol film. The

approach portion of the curve clearly shows two features of interest: a vertical contact region, where the tip reaches the rigid substrate, at $z_{tip} = 0$; and a jump-in approximately 3 µm away from the substrate. This force curve proves the presence of a 3 µm glycerol film on the substrate, and, more importantly, shows that the capillary (or frustrated capillary) force between the tip and the film is strong enough to cause a jump-in. Thus, this experiment shows that the tip is sufficiently hydrophilic to show a jump-in due to the capillary force between a liquid (or quasi-liquid) film and the tip. This means that the apparent

absence of the jump-in in most of the experiments over ice with hydrophobic tips is due to a QLL thickness below the limit of detection of this experimental technique. Considering the inherent noise in the deflection signal in these experiments, we can conclude that the QLL thickness is below 1 nm. These results are comparable or even lower than the smallest experimental QLL thicknesses previously reported (Elbaum et al., 1993; Lied et al., 1994; Dosch et al., 1995 and 1996; Bluhm et al, 2002; Sadtchenko and Ewing, 2002 and 2003), and are in the same range of computer simulation results

(Furukawa and Nada, 1997; Limmer and Chandler, 2002; Conde et al, 2008).

**4. Conclusions**

We present new results of AFM force curves over pure ice at different temperatures, performed with two different environmental chambers and different kind of AFM tips. Our results provide insight to resolve the controversy on the interpretation of experimental AFM curves. (Petrenko, 1997; Döppenschmidt and Butt, 2000; Pittenger et al., 2001)

Moreover, using silicon tips and an Enviromental Chamber (EC) with low control of the temperature of the gases in contact with the sample, we obtained force curves with jump-in distances comparable (in the lower bound) with bibliography results. On the other hand, we prove that the use of the Mini Environmental Chamber (mEC), that provides a better control of the





temperature and humidity of gases in contact with the sample, changes qualitatively the results of the experiments. This allowed us, for the first time, to get force curves over the ice-air interface with no jump-in, for some of the experiments with silicon or silicon nitride tips. These results suggest a QLL thickness below 1 nm for the explored temperature range (-7 °C to -2 °C). This upper bound is significantly lower than some of the previous AFM and IFM results (Petrenko, 1997;

Döppenschmidt and Butt, 2000; Goertz et al., 2009), which suggests that those authors overestimate equilibrium QLL thickness, due to temperature gradients or indentation of ice during the jump-in (Pittenger et al., 2001). Additionally, we proved that more hydrophobic tips (platinum covered or silanized silicon tips) render consistent force curves with no jump-in, showing that the chemistry of the tip is very relevant for study the ice-vapor interface.

Overall, this work shows that AFM measurements of the QLL thickness can be consistent with the lower range of

experimental QLL measurements from different techniques (Elbaum et al., 1993; Lied et al., 1994; Dosch et al., 1995 and 1996; Bluhm et al, 2002; Sadtchenko and Ewing, 2002 and 2003), if one chooses properly the experimental conditions (especially, the temperature and relative humidity of air and the chemistry of the tip).

This allows constraining the QLL thicknesses values in Fig. 1, which can be of significant relevance to validate QLL theories, and for multiphase atmospheric chemistry models (especially for snow-atmosphere interactions in polar regions,

glaciers, etc.). Nevertheless, it should be remarked that the effect of the QLL thickness on the atmospheric reactions is far from being completely understood. For instance, Michalowski et al. (Michalowski et al, 2000) used a multiphase model containing a large number of gas phase reactions, photolysis reactions and aqueous reactions in suspended aerosol particles and the quasi-liquid component of snow. Their model predicts much faster ozone depletion when the thickness of the QLL estimated by Conklin and Bales (Conklin and Bales 1993) is reduced by a factor 10. This is an example of a system where a

thinner QLL, as proposed in our work, would have a dramatic effect on the modeled ozone depletion.

On the contrary, McNeill et al. (McNeill et al. 2007) concluded that the HCl adsorption and surface-to-bulk flux on polar stratospheric cloud ice particles is slightly influenced by the QLL thickness, which is allowed to vary between 1 nm and 300 nm.

A good test for our claim of a low range of QLL thicknesses could be the modeling of the photochemistry of nitrate in

snowpack at temperatures in the range 250-265 K using the multiphase model by Boxe and Saiz-Lopez (Boxe and Saiz-Lopez, 2008). The authors used a QLL thickness of 300 nm that seems to be an overestimated value at the light of our results.

**Author contribution**

HRC, MPL and JGC designed AFM experiments. JGC carried out the AFM experiments, with collaboration from MPL and MMG. MMG contributed to SEM experiments. JGC, MPL, and HRC prepared the manuscript with contributions by MMG.




**Competing interest**

The authors declare that they have no conflict of interest.

**Acknowledgements**

JGC, MPL and HRC are members of Consejo Nacional de Investigaciones Científicas y Técnicas (CONICET). JGC thanks a
Fulbright / Bunge & Born grant. MMG thanks fellowships by ANPCyT and CONICET. The authors thank Dr. Andrés
Zelcer (CIBION-CONICET), for the functionalization of the AFM tip, and Dr. Paula Angelomé (CAC-CNEA) for helping
with the spin-coater.

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



**Table 1.** Average jump-in distances ($d_{jump-in}$) obtained from AFM force curves over ice with the EC using silicon tips. The dispersion ($\sigma$) obtained for various repeated measurements and the indentation slopes corresponding to the first 50 nm, are also reported. Different rows at the same temperature correspond to different (x,y) positions on the ice sample. The spring constants for the cantilevers measured at ambient temperature are also informed.

| $T$ (°C) | $d_{jump-in}$ (nm) | $\sigma$ (nm) | Indentation slope (nN/nm) | $K$ (N/m) |
|---|---|---|---|---|
| -9.5 | 7.1 | 0.4 | 1.9 | 0.087 |
|  | 6.1 | 0.8 | 3.4 |  |
| -5.0 | 6.2 | 0.3 | 7.7 | 0.12 |
|  | 5.7 | 0.4 | 4.6 |  |
|  | 6.0 | 0.4 | 5.7 |  |
| -3.5 | 4.9 | 0.3 | 0.69 | 0.079 |
|  | 8.8 | 0.9 | 0.95 |  |
|  | 8.0 | 0.7 | 0.65 |  |
| -2.0 | 37 | 3 | 0.096 | 0.057 |
|  | 5.5 | 0.4 | 0.081 |  |
|  | 14.3 | 0.3 | 0.087 |  |
| -1.0 | 8.7 | 1.3 | 1.5 | 0.079 |
|  | 5.5 | 0.2 | 2.4 |  |



**Table 2.** Average jump-in distances (and their dispersions, $\sigma$) obtained from AFM force curves over ice with the mEC for different tips (see the Supplement). The numbers of the tips correspond to different tips of equal characteristics, while different rows at the same $T_{cooler}$ represent force-curves measured on different positions over the ice surface. The indentation slopes corresponding to the first 50 nm and the spring constant for the cantilevers at the different studied temperatures are also reported.

| $T_{ice}$ (ºC) | $T_{cooler}$(ºC) | $d_{jump-in}$ (nm) | $\sigma$ (nm) | Indentation slope (nN/nm) | $K$ (N/m) | Tip |
|---|---|---|---|---|---|---|
| -3.0 | 1 to 2 | ND | | 0.15 | 0.06 | SNL1 |
| | | ND | | 0.24 | | |
| -5.0 | -1 to 0 | SUM | | SUM | | SNL2 |
| -7.0 | -1 to 0 | SUM | | SUM | | SNL3 |
| -5.0 | -0.5 to 0.0 | ND | | 0.021 | 0.06 | SNL3 |
| -5.0 | -0.2 to -0.5 | 19.0 | 0.9 | 0.17 | 0.06 | DNP1 |
| | | 13.9 | 0.9 | 0.39 | | |
| | | 17.0 | 0.6 | 5.0 | | |
| | 5.3 to 5.8 | 22 | 3 | 0.067 | | |
| | | 13 | 2 | 0.028 | | |
| | | 14 | 6 | 0.062 | | |
| -7.0 | -1.7 to -1.2 | SUM | | SUM | 0.06 | DNP2 |
| -3.0 | -0.2 to 0.8 | SUM | | SUM | 0.06 | DNP3 |
| -2.0 | 1.8 to 2.3 | ND | | 0.38 | 0.06 | DNP4 |
| | | ND | | 0.29 | | |
| -2.0 | 1.8 to 2.0 | SUM | | SUM | 0.06 | DNP5 |
| -2.0 | 2.0 to 2.8 | ND | | N | 0.06 | DNP5 |
| | | ND | | N | | |
| | | ND | | N | | |
| -3.0 | 0.3 to 0.5 | SUM | | SUM | 0.06 | DNP6 |
| -4.0 | -1.5 to -1.0 | SUM | | SUM | 0.06 | |
| -4.0 | -1.0 to -0.5 | SUM | | SUM | 0.06 | SNP7 |





| -6.0 | -2 to -1.5 | SUM | | SUM | | |
|------|-----------|-----|---|-----|------|------|
| -4.0 | 0 to 0.3 | ND | | 0.066 | 0.21 | PIC1 |
| | | 14 | 1 | 0.045 | | |
| | | 16 | 2 | 0.076 | | |
| | | 8 | 2 | 0.033 | | |
| | | 10 | 1 | 0.042 | | |
| -2.0 | 0.3 to 0.5 | SUM | | SUM | 0.21 | PIC1 |
| -3.0 | | SUM | | SUM | | |
| -6.5 | -0.8 to -0.5 | ND | | 0.28 | 0.23 | PIC2 |
| | | ND | | 0.25 | | |
| -2.0 | 1.5 to 1.8 | ND | | 0.15 | 0.23 | |
| | | ND | | 0.11 | | |
| | | ND | | 0.14 | | |
| | | ND | | 0.12 | | |
| -4.0 | 1.5 to 2.0 | ND | | 0.14 | 0.23 | PIC3 |
| -2.0 | 1.5 to 2.0 | ND | | 0.083 | | |
| -3.0 | 0.0 to 0.3 | ND | | 0.47 | 0.15 | PIC5 |
| | | ND | | 0.3 | | |
| -2.0 | -1.5 to -1.2 | ND | | 0.089 | 0.12 | DNP-S1 |
| | | ND | | 0.090 | | |

\* SUM corresponds to measurements for which the signal in the laser detector was lost, ND corresponds to measurements where the jump-in was not detected, N corresponds to measurements which could not be quantified due to the noise in the determination.



**Table 3.** EDS analysis of the Pt/Ir (PIC) tips, before and after usage.

| Element | Wt% | Wt% |
|---|---|---|
| | New tip | Used tip |
| C | 23.18 | 27.21 |
| O | 6.65 | 16.62 |
| Al | 4.96 | 7.38 |
| Si | 45.11 | 32.70 |
| Cu | 0.63 | 0.73 |
| Pt | 19.47 | 10.20 |
| S | | 0.37 |
| Cl | | 0.45 |
| Zn | | 4.34 |





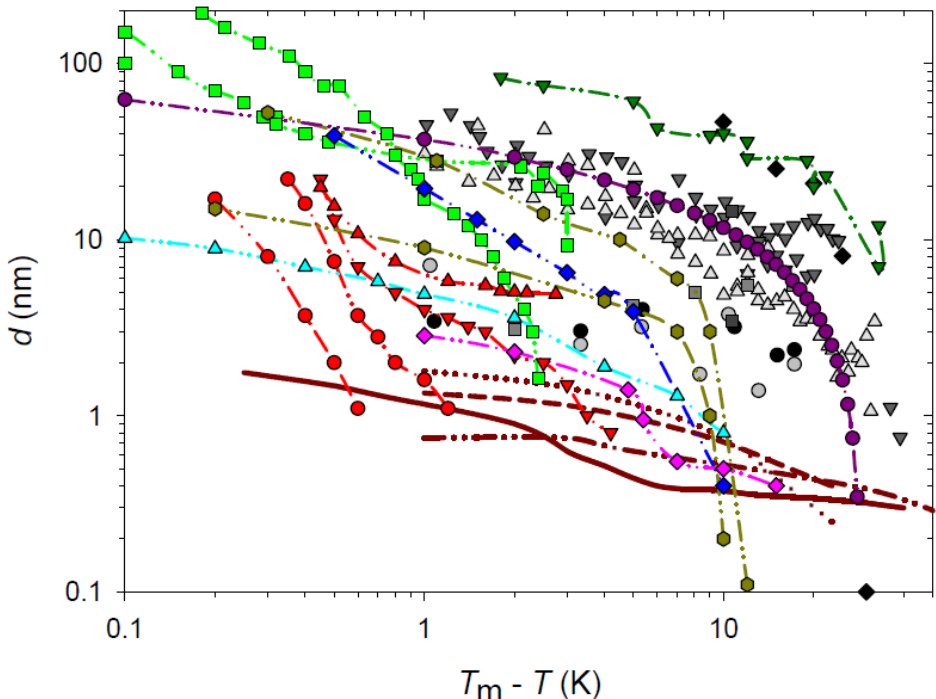

**Figure 1: QLL thickness determined with different experimental and simulation techniques: AFM measurements by Pittenger et al. (Pittenger et al., 2001) with Si tips (◯) and hydrophobic coated silicon tips (●); by Döppenschmidt et al. (Döppenschmidt et al., 2000) in air (▼) and vacuum (△); by Petrenko (Petrenko, 1997) (■); by Bluhm et al. (BluhmandSalmeron, 1999 and Bluhm et al., 2000) (▨). IFM measurements by Goertz et al. (Goertz et al., 2009) (◆). Brewster reflectometry by Elbaum et al. (Elbaum et al., 1993) (●▲▼; different symbols correspond to different experiments). Ellipsometry by Beaglehole and Nason (Beaglehole and Nason, 1980) (◆), and Furukawa et al. (Furukawa et al., 1987) (▧). XPS by Bluhm et al. (Bluhm et al., 2002) (◆). FTIR by Sadtchenko and Ewing (Sadtchenko and Ewing, 2002) (▲). Proton dispersion by Golecki and Jaccard (Goleki and Jaccard, 1977) (▼). GXRD by Dosch et al. (Dosch et al., 1995 and 1996), and Lied et al. (Lied et al., 1994) (●). Simulation results by Limmer and Chandler (Limmer and Chandler, 2002) (——); Furukawa and Nada (Furukawa and Nada, 1997) (⋯⋯⋅ and ━ ━ ━) for the prismatic and basal planes, respectively, and by Conde et al. (Conde et al., 2008) (——ᐧ ᐧ) for the basal plane.**



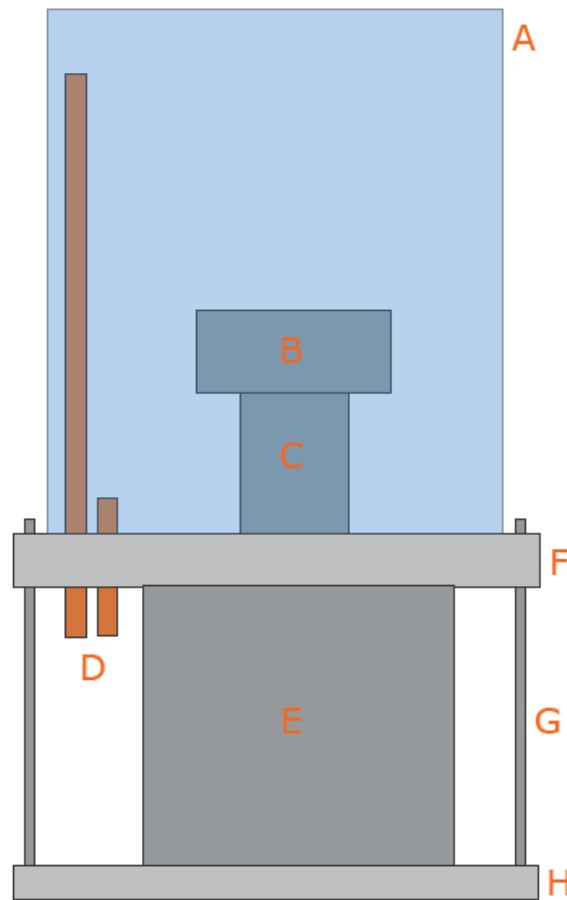

**Figure 2: Scheme of the EC: (A) acrylic chamber, (B) AFM head, (C) piezoelectric tube, (D) copper tubes (inlet and outlet of gases), (E) AFM base, (F) aluminum ring, (G) threaded rods, (H) aluminum base.**



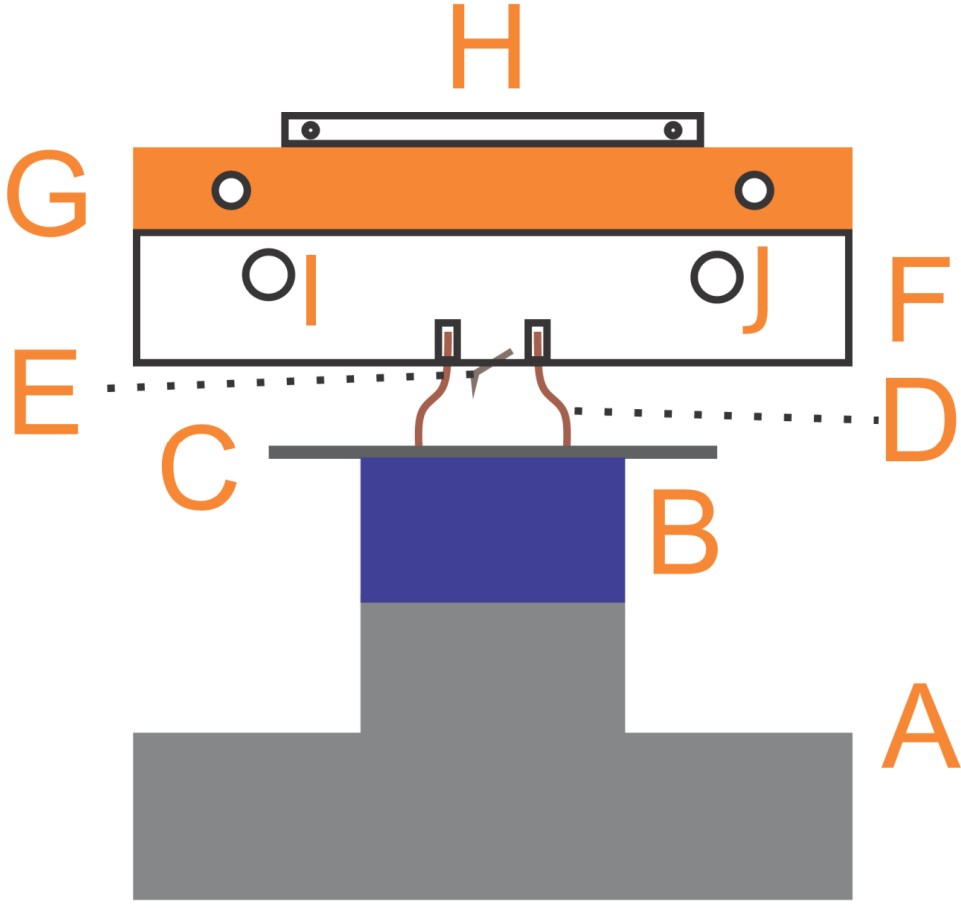

**Figure 3: Scheme of the front view of the AFM and the mEC: (A) piezoelectric, (B) Peltier element, (C) mica substrate, (D) silicone o-ring, (E) AFM tip, (F) fluid´s glass cell, (G) copper cooler, (H) heating resistance, (I) outlet of gases and temperature-humidity sensor, (J) inlet of gases.**





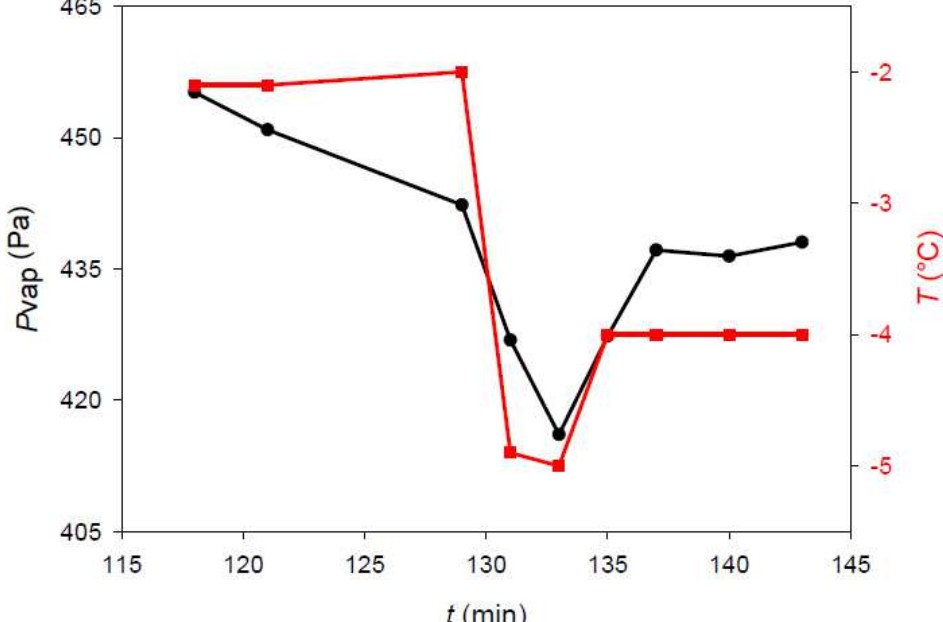

**Figure 4: Typical behavior of the humidity during ice deposition on mica in the mEC: (●) Water vapor pressure in the chamber, (■) Substrate temperature.**





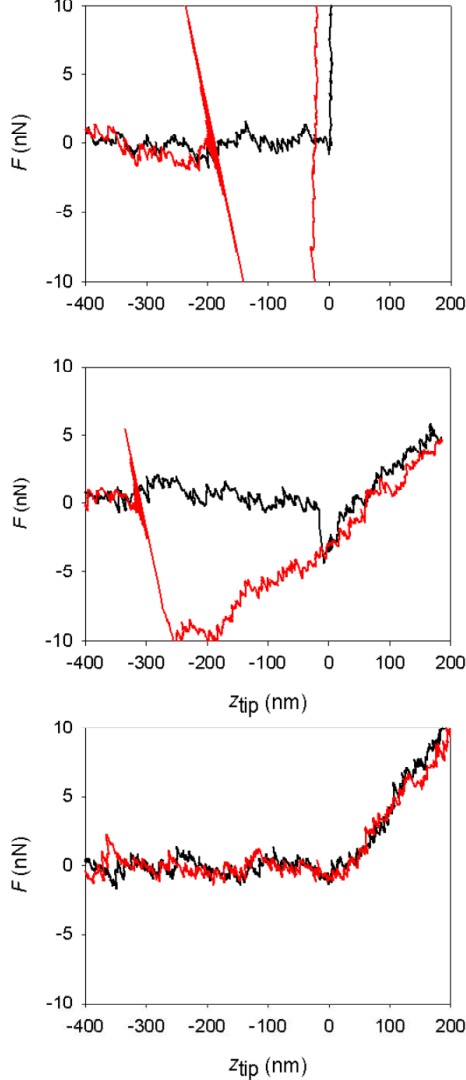

**Figure 5: Different shapes of AFM force curves. Top panel (a): mica; center (b) and bottom (c) panels: ice. Curves in black represent the approach branches and curves in red the retract branches.**





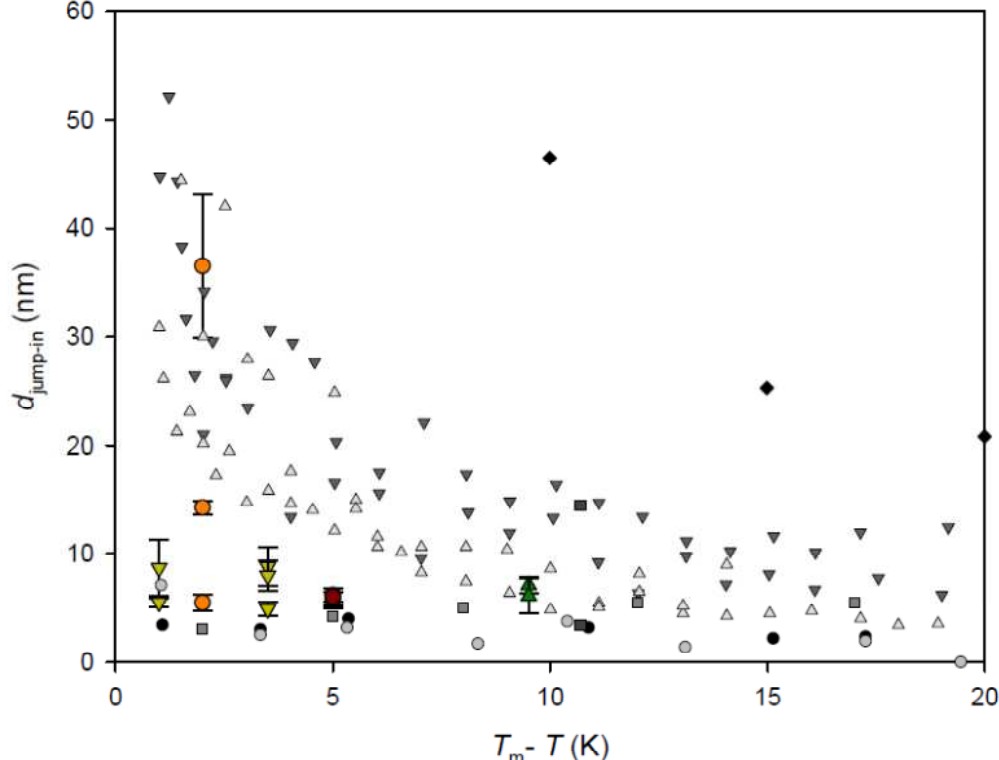

**Figure 6: AFM jump-in distances obtained with the EC, in color symbols. For comparison, results of QLL thicknesses obtained from literature with similar techniques are plotted in the same figure. Symbols for results by other authors are the same as in Fig. 1.**



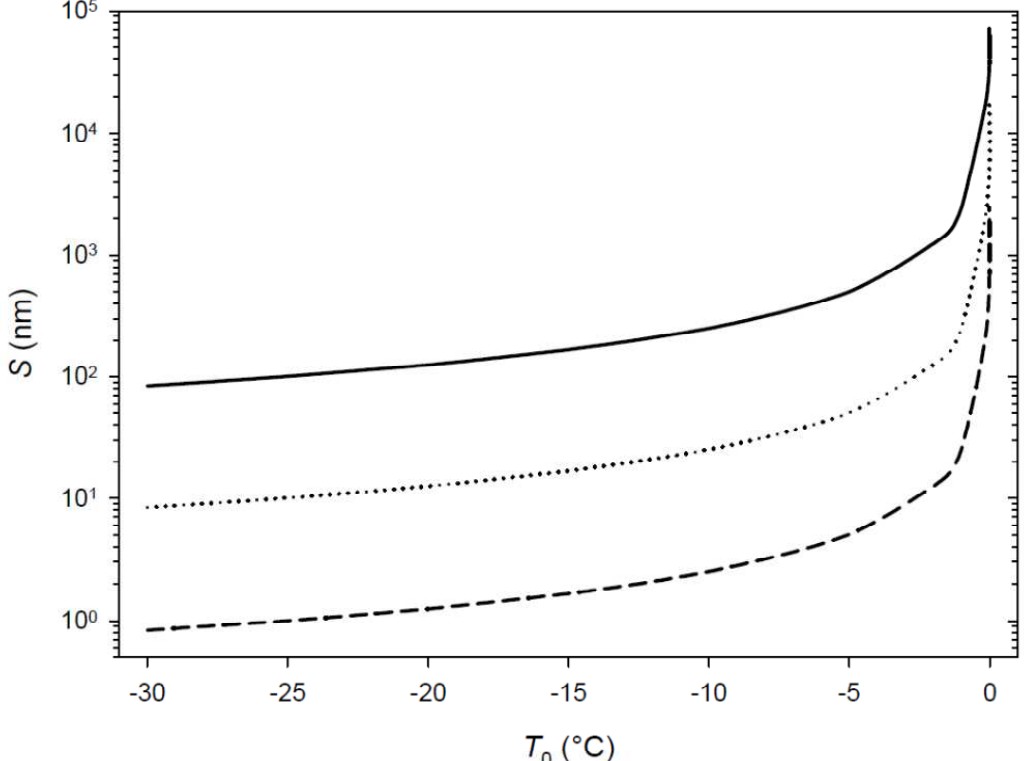

**Figure 7: Thickness of the liquid layer over ice in presence of a temperature gradient. Total thickness (ice + liquid water): 100 μm. Full line: $T_L$ = 0.1 °C. Dotted line: $T_L$ = 0.01 °C. Dashed line: $T_L$ = 0.001 °C.**





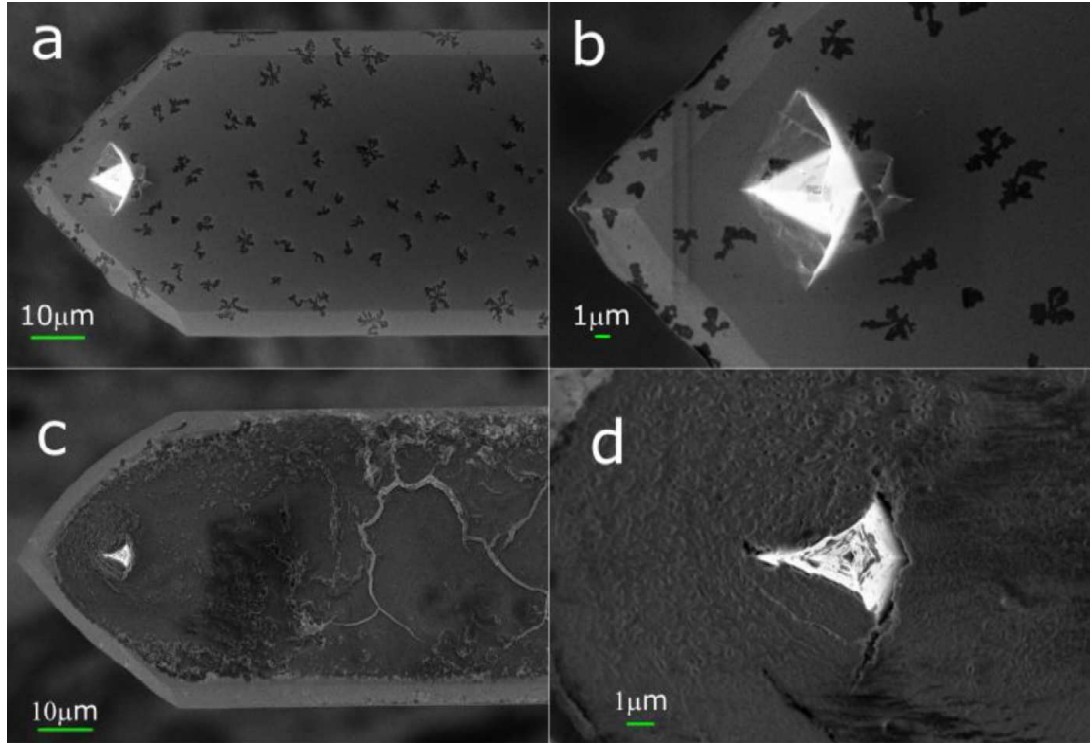

**Figure 8: SEM Micrographs of the PIC1 AFM tip. The upper panels correspond to the micrographs of the tip before usage and the lower panels to those of the tip after usage. Panels a) and c) correspond to a magnification of 3.00 KX and panels b) and d) to magnifications of 8.00 KX and 15.00 KX, respectively.**




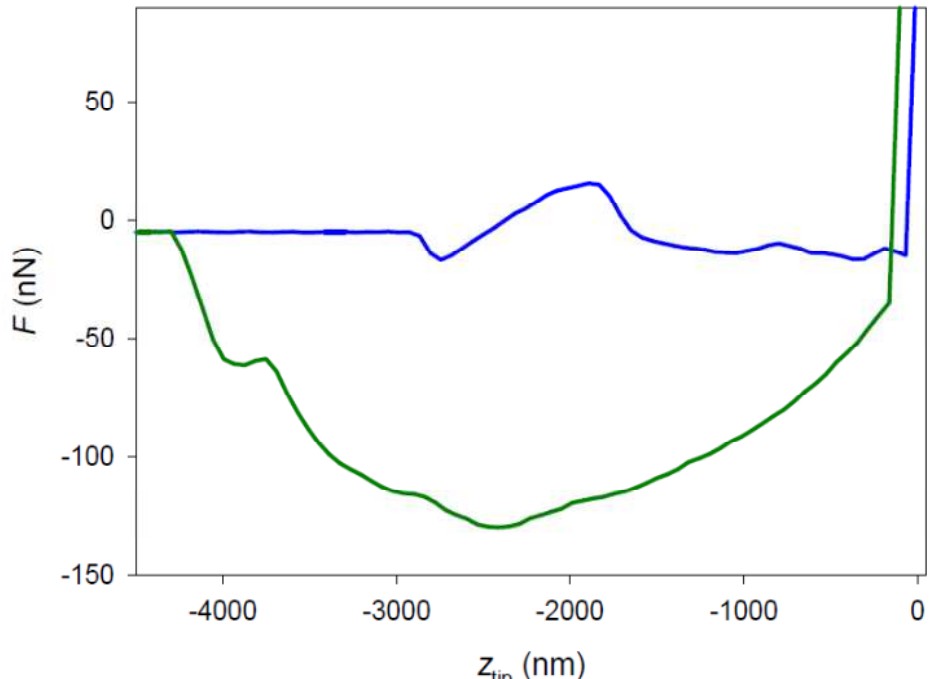

**Figure 9: AFM force curve (approach (blue), and retract (green)) performed with a platinum covered tip over a thin glycerol film deposited over silicon at a temperature of 5 ºC, using the mEC.**

