# Peer review of "The Quasi-Liquid Layer of ice revisited: the role of temperature gradients and tip chemistry in AFM studies"

_Atmospheric Chemistry and Physics, 2017_

## Referee Comment (RC1) · Anonymous Referee #1 · 6 Jul 2018

This paper presents AFM results on the ice-air interface in order to determine the thickness of the quasi-liquid layer which is an important question for understanding amongst others atmospheric processes. The authors overview the current literature very well and show that there is a wide variety of QLL thicknesses obtained with AFM. In the current manuscript the authors describe AFM experiments performed in a very small chamber with controlled temperature and humidity. In this way they show that temperature gradients are very important. Small temperature gradients yield a small QLL thickness. Moreover, the hydrophobicity of the tip is playing a role. Based on their results, the inconsistency in literature data could be explained. The paper is well written and the conclusions are well supported by the data. After considering the comments

below, I recommend publication in ACP.

Specific comments:

1. From the experimental part it is not clear what type of ice is used. Is it amorphous or crystalline? If crystalline, what is the surface phase? What is the surface structure?

2. Along the same lines; in overviewing the literature in the text and in Fig. 1 the type of ice is not mentioned. The authors should elaborate a bit on it.

3. For non-AFM experts like me, it would be very helpful to connect Fig. 5 and Table 1 by marking in the figure the d_jump-in and the indentation slope.

4. The caption of Fig. 5 is missing important information. It should make clear what the difference between the center and bottom panel is. It is apparent from the text, but not from just reading the figure caption.
* * *

---

## Author Comment (AC2) · 17 Sep 2018

The authors deeply thank the appreciative comments by Referee #2, Thorsten Bartels-Rausch, and his editorial decision to accept the manuscript for final publication in ACP after dealing with his comments.

Regarding the specific questions:

1. Additionally, I'd ask you to comment on the following point and modify the manuscript accordingly. It is not entirely clear to me how the AFM measurement are calibrated relative to the ice sample thickness. If I understood correctly, the distance of the solid ice

is z=0, How is that determined? In other words, measuring the force-distance how do you determine the thickness of the QLL without determining the samples topography, or thickness. It appears to me that this calibration is a central issue in AFM and I would ask you to comment on this in more detail.

Answer: Indeed, calibration is a central issue in AFM indentation experiments. We decided to give a detailed discussion of calibration in the Supplement, in order not to disturb the reading of the main article by non-AFM experts. We believe that the most original contributions in our work are the improvements regarding temperature gradients and the chemistry of the tip. We did not want to distract the attention from those aspects. Nevertheless, we do discuss in the main article some of the assumptions needed for the interpretation of force curves and calibration mistakes that would lead to overestimation of the QLL thickness (pages 8-9 of this manuscript). Regarding the specific doubts presented by the Referee, first, it must be clarified that the measured jump-in distance (and, hence, the QLL thickness) does not depend on the zero distance definition. This issue is relevant in the conversion of the force curves after contact, hence it must be taken into account for studies of indentation, viscoelastic models, etc. We added a paragraph in the Supplement which explains that the zero distance definition does not affect the QLL thicknesses determined in this work, and we also added two references that discuss these calibration issues in detail. In the same way, the ice samples thickness does not have an impact on QLL thickness determination, since AFM force-curves study the ice-QLL surface. We did find relevant to add a paragraph and a figure (Fig. S6) clarifying that we do have a lower bound of ice thicknesses, which are large enough to discard influence of the substrate and nano-confinement effects. The same is not true for results by Bluhm et al., as the Referee remarked in another comment (see below).

Changes in the manuscript: Page 7, lines 20-29: We did not observe the ice droplets mentioned by Bluhm and coworkers, (Bluhm et al., 2000), probably due to the fact that they prepared ice samples with thicknesses of few ice bilayers. In our case, ice samples

thicknesses were not measured systematically (as it was not the focus of this work), but in most of the experiments ice thickness exceeded the spanned indentation depth (hundreds of nanometers to 5 $\mu$m, the maximum vertical distance that can be measured with the AFM scanner used for these experiments). In few experiments we were able to measure the ice thickness, since the AFM tip reached the infinitely hard mica substrate (Fig. S6, Supplement). Considering these macroscopic thicknesses, we can assure that we studied the ice-vapor interface without the influence of the underlying substrate or nano-confinement effects.

Supplement, page 3, lines 16-19: It must be emphasized that one of the main parameters determined in this work, the jump-in distance, is calculated as a difference between ztip distances: $d_{(jump-in)} = z\_tip^C - z\_tip^B$ , where superscript B and C refer to positions of the tip at Fig. S1. Hence, the jump in distance is directly influenced by calibrations of zpiezo and Sens (from eq. S.5), but does not depend on the calibration of the spring contact or the zero distance definition.

Supplement, page 3, line 33 – page 4, lines 1-3: Figure S6 shows one of the few cases when we could measure explicitly the thickness of the ice sample. The thickness can be estimated from the difference between the end of jump-in (ztip = 0) and the infinite slope (mica substrate, ztip $\approx$ 40 nm). In most of the experiments, thicknesses were much larger, therefore the tip did not reach the mica substrate. In those cases, we can only infer a lower bound for the ice thickness (the maximum indentation depth).

2. Small comments Page 1, Line 25: melted layer -> pre-melted layer

Answer: We thank the Referee for noting this inconsistency in nomenclature. The use of the term "premelting" and "premelted layer" has been discussed, as it might suggest a direct link between the QLL and true melting. Our discussion takes that difference into account, but in some parts of the article we have been inconsistent in the language.

Changes in the manuscript: Page 1, Line 25: Slightly below the melting temperature, Tm, a disordered layer in the solid-vapor interface has been observed in many crystalline solids. This layer is commonly called in the literature "quasi-liquid layer" (QLL), since many of its properties differ from those corresponding to the bulk supercooled liquid at the same temperature.

Page 2, Line 13: The physics of the disordered surface in ice and its geophysical consequences have been reviewed by Dash et al. (Dash et al, 2006) and Bartels-Rausch et al. (Bartels-Rausch et al, 2014), who reported a comparison between calculated and measured QLL thicknesses.

3. Page 9, Line 6: Apparently, Bluhm probed nm thick ice growing on a support. I would argue that the structure of that ice does not reflect the surface of bulk ice crystals, but rather of nano-films on a support being influence on that support. So, I would suggest to highlight more specifically that this study did not probe the QLL on ice.

Answer: We thank the reviewer for the suggestion. We extended the paragraph regarding Bluhm and coworkers experiments to highlight this fact.

Changes in the manuscript: Page 9, Line 31: Results by Bluhm et al. (Bluhm and Salmeron, 1999; Bluhm et al., 2000) present an opposite trend to most of the literature values, that is, lower QLL thicknesses at higher temperatures. However, it should be noted that they measure QLL thickness over much thinner samples (0.3 nm to 3 nm), which corresponds to a few bilayers of ice-like water molecules on the substrate. It should be stressed that these experiments do not give information on the QLL of bulk ice, even if the structure of water on the substrate could be related to that of crystalline ice. Nano-films properties might show a large dependence on the thickness and the influence of the substrate.

Please also note the supplement to this comment:
https://www.atmos-chem-phys-discuss.net/acp-2017-1213/acp-2017-1213-AC2-supplement.pdf

2018.

**Supplement:**

[revised manuscript text omitted]

---

## Author Comment (AC3)

[revised manuscript text omitted]
_{\text{tip}}/\text{nm} = z_{\text{piezo}}/\text{nm} - \text{Defl}/\text{nm} = z_{\text{piezo}}/\text{nm} - \frac{\text{Defl/V}}{\text{Sens/(V/nm)}} \qquad (\text{S.5})$$

5    We normalized the curves by fixing $z_{\text{tip}} = 0$ at position C in Fig.1S, under the assumption that the contact region begins at that distance. This implies suppositions (Butt, 2005; Attard, 2007) that are analyzed in the paper text.

Since laser alignment might change during the experiments, which would affect deflection measurements, an additional calibration was performed. After deposition of ice, which does not behave as an "infinitely hard" substrate (Butt et al., 2000; Pittenger et al., 2001), there is no possibility to check if the deflection sensitivity has changed. D'Costa and Hoh (D'Costa

10   and Hoh, 1995) performed several calibrations of the deflection sensitivity on a hard substrate, for slightly different alignments of the laser beam, while keeping the same effective length of the cantilever. They found a linear relationship between the deflection sensitivity, and the shift of the potential (V) for a fixed displacement of the detector in $z$ direction. Figure S2 shows the results of a similar calibration for our AFM. During the experiments, we retracted the tip periodically and checked for modifications in V. In very few cases, we observed a change in V (due to a small change in the laser beam

15   alignment), and in those cases we used the calibration curve to recalculate Sens.

It must be emphasized that one of the main parameters determined in this work, the jump-in distance, is calculated as a difference between $z_{\text{tip}}$ distances: $d_{jump-in} = z_{tip}^{C} - z_{tip}^{B}$, where superscript B and C refer to positions of the tip at Fig. S1. Hence, the jump in distance is directly influenced by calibrations of $z_{\text{piezo}}$ and *Sens* (from eq. S.5), but does not depend on the calibration of the spring contact or the zero distance definition.

20

**S2. AFM Force Curves**

The following figures represent some of the experimental AFM force curves from which results in Table 2 of the main article have been extracted.

Figures S3 and S4 represent continuous force curves measurements performed with silicon nitride tips in the same position

25   over the ice surface, which show high reproducibility. The reproducibility of all the force curves can be captured by considering the averages and standard deviations of their key features (ice indentation slope and jump-in distance) on Table 2.

Figure S5 shows the approach and retract portions of sample force curves measured on ice with platinum covered silicon nitride tips. It can be seen that for these curves the approach portion does not present jump-in, and the retract portion does

30   not presents adhesion (opposite to the large adhesion forces represented in the upper and central panels of Figure 5 in the main article).As we describe in the main article, most of the AFM force curves measured with platinum covered tips or silanized tips on the mEC share these features.

Figure S6 shows one of the few cases when we could measure explicitly the thickness of the ice sample. The thickness canbe

estimated from the difference between the end of the jump-in ($z_{tip} = 0$) and the infinite slope (mica substrate, $z_{tip} \approx 40$ nm). In most of the experiments, ice thicknesses were much larger, therefore the tip did not reach the mica substrate. In those cases, we can only infer a lower bound for the ice thickness (the maximum indentation depth).

25

30

**Table S1:** Nominal parameters of the commercial AFM tips (Bruker/Veeco). $R$: curvature radii of the tip. $K$: force constant, $\nu_0$: resonance frequency and $\tau$: period of the cantilever. For the curvature radii ($R$), maximum values reported by the manufacturer areindicated in brackets.

| Material | Model | $R$ (nm) | $K$ (N/m) | $\nu_0$ (kHz) | $\tau$ (µs) |
|----------|-------|----------|-----------|---------------|-------------|
| Si | SNL | 2 (max 12) | 0.06-0.35 | 18-65 | 15-56 |
| $Si_3N_4$ | DNP | 20 (max 60) | 0.06-0.35 | 18-65 | 15-56 |
| Si, Pt/Ir | SCM-PIC | 20 (max 25) | 0.2 | 13 | 77 |

15

[Figure]

**Figure S1: Typical AFM force curves, (approach ~O,~ and retract ()). ~Left~ panel represents and infinitely hard substrate, while central panel represents a soft sample. The diagrams on the right panel show the tip position and cantilever flexion for different distances indicated with letters on the force curves. The red line in diagram D.i (hard substrate) represents the deflection distance. The red line in diagram D.ii (soft sample) represents the indentation depth.**

[Figure]

**Figure S2:Example of a sensibility (Sens) calibration curve as a function ofvoltage (ΔV), performed with a platinum tip on mica at 4$^0$C.**

[Figure]

**Figure S3: AFM force curves (with jump-in) measured on the same position on the ice surface at -5.0 ºC with a silicon nitride tip and the mEC.**

[Figure]

**Figure S4: AFM force curves (whithout jump-in) measured on the same position on the ice surface at -2.0 °C with a silicon nitride tip and the mEC.**

[Figure]

**Figure S5: AFM force curves measured with platinum covered silicon nitride tips in the mEC. Top left: -6.5 ºC. Top right: -4.0 ºC. Bottom left: -3.0 ºC. Bottom right: -2.0 ºC. Black lines represent the approach portion of the force curves, whereas the red lines correspond to the retract portion.**

[Figure]

Figure S6:AFM force curve measured at -2.0 °C with a silicon nitride tip and the mEC. Ice sample thickness can be estimated as the difference between the end of jump-in ($z_{tip}$ = 0) and the infinite slope (mica substrate, $z_{tip} \approx$ 40nm). The black curve corresponds to the approach portion of the force curve and the red one to the retract portion.

---

## Author Response (AR1)

**Interactive comment AC1 on "The Quasi-Liquid Layer of ice revisited: the role of temperature gradients and tip chemistry in AFM studies" by Julián Gelman Constantin et al.**

**Reply to RC1: 'Referee Comment to acp-2017-1213', Anonymous Referee #1, 06 Jul 2018**

The authors deeply thank the appreciative comments by Anonymous Referee #1. Regarding the specific questions:

**1. From the experimental part it is not clear what type of ice is used. Is it amorphous or crystalline? If crystalline, what is the surface phase? What is the surface structure?**

**Answer:** We did not intend to prepare ice single crystals and study specific face orientations, since these are not the kind of samples expected to find in environmental conditions. Our preparation method (similar to that of other experiment) surely produces polycrystalline ice. As we detail in section 2.4, water vapor was deposited at slow rates (oversaturation lower than 120 %), and the ice layer was stabilized for 10 to 20 minutes prior to measurement. Deposition conditions are close enough to ice-vapor equilibrium conditions to ensure ice crystals are formed (instead of amorphous ice). Stabilization time is enough to allow ice crystals to reach equilibrium or near-equilibrium conditions (when vapor pressure is controlled).

**Changes in the manuscript:**
**Page 7, line 7:** Even though we did not perform specific experiments to verify the structure and orientation of the prepared ice, we claim that we studied polycrystalline ice. This procedure is rather similar to that of many of the reviewed experiments (Bluhm et al., 2000; Bluhm et al., 2002; Pittenger et al., 2001). Pittenger and coworkers (Pittenger et al., 2001) report to obtain polycrystalline ice with its surface smooth at the scale visible to optical microscope. In some of the experiments, we obtained AFM contact images that confirmed a smooth ice surface (roughness lower than 5 nm in most cases, images not shown). We did not observe the ice droplets mentioned by Bluhm and coworkers, (Bluhm et al., 2000), probably due to the fact that they prepared much thinner ice layers. Bluhm and coworkers claim thicknesses of few ice bilayers, whereas in most of our experiments we studied layers thicker than 5 µm (the largest measurable depth in our AFM).

**2. Along the same lines; in overviewing the literature in the text and in Fig. 1 the type of ice is not mentioned. The authors should elaborate a bit on it.**

**Answer:** We thank the referee for the remark. Indeed, differences in ice sample preparation are possibly another contribution to the large discrepancies between different experiments. We remarked that on the manuscript. We also add a new reference regarding this topic (an article based on a conference proceeding cited in the former version of this manuscript).

**Changes in the manuscript:**

**Page 2, Line 32:** AFM experiments, for instance, involve the interaction of a tip with the sample; thus, it is uncertain whether other phenomena are also involved in the measurements. About half of the reviewed experiments report special procedures to prepare ice single crystals, and specified the studied crystal face or faces (Petrenko, 1997; Beaglehole and Nason, 1980; Furukawa et al., 1987; Elbaum et al, 1993; Dosch et al., 1995 and 1996; Lied et al., 1994; Goleki and Jaccard, 1977). The remaining authors report simpler ice sample preparation techniques, which very likely produce polycrystalline ice (Doppenschmidt and Butt, 2000; Bluhm et al, 2000; Pittenger et al., 2001; Goertz et al., 2009; Bluhm et al., 2002; Sadtchenko and Ewing, 2002) or, in one case, single crystals with unknown orientation (Doppenschmidt and Butt, 2000). The effect of the crystal face on the QLL thickness was found to be relevant in some of the experiments, but results from different experiments give in some cases opposite relation between the thickness of basal and primary prismatic plane (Beaglehole and Nason, 1980; Furukawa et al., 1987; Dosch et al., 1995). Molecular dynamic simulations give more subtle differences between cristal faces (Gelman Constantin et al., 2015; Pickering et al., 2018).

**Page 3, Line 27:** However, further simulations by Gelman Constantin (Gelman Constantin, 2015b) show that hydrophilic tips can induce frustrated capillarity between the tip and the QLL. Recent simulations by Pickering and coworkers (Pickering et al., 2018) showed similar results.

**Page 8, Line 9:** Computer simulations (Gelman Constantin, 2015; Pickering et al., 2018) have also shown that for hydrophilic AFM tips the QLL deforms to reach the tip (frustrated capillarity), which would produce jump-in distances larger than equilibrium QLL thicknesses.

**Page 12, Line 2:** Thus, the later experimental conditions could lead to thicker (non-equilibrium) QLL values.  Pickering et al. (Pickering et al., 2018) performed Grand-Canonical MD simulations under condensation conditions (over-saturation) and found non-equilibrium QLL thicknesses larger than the equilibrium values. They suggest that "the pressure should be controlled in experiments with a precision of at least 10% or that it should be kept slightly below the saturation point, where the QLL depth would not be significantly affected", as we found in our experiments.

**Page 12, Line 23:** These results are comparable or even lower than the smallest experimental QLL thicknesses previously reported (Elbaum et al., 1993; Lied et al., 1994; Dosch et al., 1995 and 1996; Bluhm et al, 2002; Sadtchenko and Ewing, 2002 and 2003), and are in the same range of computer simulation results (Furukawa and Nada, 1997; Limmer and Chandler, 2002; Conde et al, 2008; Pickering et al., 2018). Subtle differences between QLL thicknesses of different crystal faces in these studies remain to be confirmed and are out of the scope of this work.

**Page 16, Line 11:**

Petrenko, V. F.: The surface of ice, USA Cold Regions Research and Engineering Laboratory Special Report; 94-22, 1994.

Petrenko, V. F.: Study of the surface of ice, ice/solid and ice/liquid interfaces with scanning force microscopy, J. Phys. Chem. B, 101, 6276-6281, 1997.

Pittenger, B., Fain, S. C., Cochran, M. J., Doney, J. M. K., Robertson, B. E., Szuchmacher, A., and Overney, R. M.: Premelting at ice-solid interfaces studied via velocity-dependent indentation with force microscope tips, Phys. Rev. B., 63, 134102, 2001.

Pickering, I., Paleico, M., Perez Sirkin, Y.A., Scherlis, D.A. and Factorovich, M.H.: Grand Canonical Investigation of the Quasi Liquid Layer of Ice: Is It Liquid? J. Phys. Chem. B, 122, 4880–4890, 2018.

**3. For non-AFM experts like me, it would be very helpful to connect Fig. 5 and Table 1 by marking in the figure the d_jump-in and the indentation slope.**

**Answer:** Thank you for your suggestion, we modified Fig. 5.

**Changes in the manuscript:**
**Figure 5:** see updated manuscript as a supplement to this comment.
**Caption of the Figure:** see next, in referee comment **#4**.

**4. The caption of Fig. 5 is missing important information. It should make clear what the difference between the center and bottom panel is. It is apparent from the text, but not from just reading the figure caption.**

**Answer:** We tried to clarify the differences between both panels, even though a caption might not be enough to capture all the differences.

**Changes in the manuscript:** Caption of Figure 5 (page 25):
 Different shapes of measured AFM force curves. Top panel (a): mica, Environmental Chamber (EC), silicon tip. Center panel (b): ice, EC, silicon tip. Bottom panel (c): ice, Micro Environmental Chamber (mEC), platinum coated silicon tip. Curves in black represent the approach branches and curves in red the retract branches. Green line marks the jump-in-distance, blue dotted line is parallel to the indentation slope.

**Interactive comment AC2 on "The Quasi-Liquid Layer of ice revisited: the role of temperature gradients and tip chemistry in AFM studies" by Julián Gelman Constantin et al.**

**Reply to RC2: 'Referee comment.', Thorsten Bartels-Rausch, 30 Aug 2018**

The authors deeply thank the appreciative comments by Referee #2, Thorsten Bartels-Rausch, and his editorial decision to accept the manuscript for final publication in ACP after dealing with his comments.

Regarding the specific questions:

**1. Additionally, I'd ask you to comment on the following point and modify the manuscript accordingly. It is not entirely clear to me how the AFM measurement are calibrated relative to the ice sample thickness. If I understood correctly, the distance of the solid ice is z=0, How is that determined? In other words, measuring the force-distance how do you determine the thickness of the QLL without determining the samples topography, or thickness. It appears to me that this calibration is a central issue in AFM and I would ask you to comment on this in more detail.**

**Answer:** Indeed, calibration is a central issue in AFM indentation experiments. We decided to give a detailed discussion of calibration in the Supplement, in order not to disturb the reading of the main article by non-AFM experts. We believe that the most original contributions in our work are the improvements regarding temperature gradients and the chemistry of the tip. We did not want to distract the attention from those aspects.

Nevertheless, we do discuss in the main article some of the assumptions needed for the interpretation of force curves and calibration mistakes that would lead to overestimation of the QLL thickness (pages 8-9 of this manuscript).

Regarding the specific doubts presented by the Referee, first, it must be clarified that the measured jump-in distance (and, hence, the QLL thickness) does not depend on the zero distance definition. This issue is relevant in the conversion of the force curves *after contact*, hence it must be taken into account for studies of indentation, viscoelastic models, etc. We added a paragraph in the Supplement which explains that the zero distance definition does not affect the QLL thicknesses determined in this work, and we also added two references that discuss these calibration issues in detail.

In the same way, the ice samples thickness does not have an impact on QLL thickness determination, since AFM force-curves study the ice-QLL surface. We did find relevant to add a paragraph and a figure (Fig. S6) clarifying that we do have a lower bound of ice thicknesses, which are large enough to discard influence of the substrate and nano-confinement effects. The same is not true for results by Bluhm et al., as the Referee remarked in another comment (see below).

**Changes in the manuscript:**
**Page 7, lines 20-29:** We did not observe the ice droplets mentioned by Bluhm and coworkers, (Bluhm et al., 2000), probably due to the fact that they prepared

and coworkers claim ice samples with thicknesses of few ice bilayers, whereas in most of our experiments we studied layers thicker than 5 µm (the largest measurable depth in our AFM).. In our case, ice samples thicknesses were not measured systematically (as it was not the focus of this work), but in most of the experiments ice thickness exceeded the spanned indentation depth (hundreds of nanometers to 5 µm, the maximum vertical distance that can be measured with the AFM scanner used for these experiments). In few experiments we were able to measure the ice thickness, since the AFM tip reached the infinitely hard mica substrate (Fig. S6, Supplement). Considering these macroscopic thicknesses, we can assure that we studied the ice-vapor interface without the influence of the underlying substrate or nano-confinement effects.

**Supplement, page 3, lines 16-19:**

It must be emphasized that one of the main parameters determined in this work, the jump-in distance, is calculated as a difference between $z_{tip}$ distances: $d_{jump-in} = z_{tip}^C - z_{tip}^B$, where superscript B and C refer to positions of the tip at Fig. S1. Hence, the jump in distance is directly influenced by calibrations of $z_{piezo}$ and *Sens* (from eq. S.5), but does not depend on the calibration of the spring contact or the zero distance definition.

**Supplement, page 3, line 33 – page 4, lines 1-3:**

Figure S6 shows one of the few cases when we could measure explicitly the thickness of the ice sample. The thickness can be estimated from the difference between the end of jump-in ($z_{tip} = 0$) and the infinite slope (mica substrate, $z_{tip} \approx 40$ nm). In most of the experiments, thicknesses were much larger, therefore the tip did not reach the mica substrate. In those cases, we can only infer a lower bound for the ice thickness (the maximum indentation depth).

**2. Small comments Page 1, Line 25: melted layer -> pre-melted layer**

**Answer:** We thank the Referee for noting this inconsistency in nomenclature. The use of the term "premelting" and "premelted layer" has been discussed, as it might suggest a direct link between the QLL and true melting. Our discussion takes that difference into account, but in some parts of the article we have been inconsistent in the language.

**Changes in the manuscript:**

**Page 1, Line 25:** Slightly below the melting temperature, $T_m$, surface premelting a disordered layer in the solid-vapor interface has been observed in many crystalline solids. This melted layer is commonly called in the literature "quasi-liquid layer" (QLL), since many of its properties differ from those corresponding to the bulk supercooled liquid at the same temperature.

**Page 2, Line 13:** The physics of the disordered surfacepremelting phenomena in ice and its geophysical consequences have been reviewed by Dash et al. (Dash et al, 2006) and Bartels-Rausch

et al. (Bartels-Rausch et al, 2014), who reported a comparison between calculated and measured QLL thicknesses.

**3. Page 9, Line 6: Apparently, Bluhm probed nm thick ice growing on a support. I would argue that the structure of that ice does not reflect the surface of bulk ice crystals, but rather of nano-films on a support being influence on that support. So, I would suggest to highlight more specifically that this study did not probe the QLL on ice.**

**Answer:** We thank the reviewer for the suggestion. We extended the paragraph regarding Bluhm and coworkers experiments to highlight this fact.

**Changes in the manuscript:**

**Page 9, Line 31:**

[revised manuscript text omitted]